# MET Oncogene Enhances Pro-Migratory Functions by Counteracting NMDAR2B Cleavage

**DOI:** 10.3390/cells13010028

**Published:** 2023-12-21

**Authors:** Simona Gallo, Annapia Vitacolonna, Paolo Maria Comoglio, Tiziana Crepaldi

**Affiliations:** 1Department of Oncology, University of Turin, Regione Gonzole 10, 10143 Orbassano, Italy; simona.gallo@unito.it (S.G.); annapia.vitacolonna@unito.it (A.V.); 2Candiolo Cancer Institute, FPO-IRCCS, SP142, Km 3.95, 10060 Candiolo, Italy; 3IFOM ETS—The AIRC Institute of Molecular Oncology, Via Adamello 16, 20139 Milano, Italy; pcomoglio@gmail.com

**Keywords:** MET tyrosine kinase receptor, glutamate receptor, NMDAR, cancers, proteolytic cleavage, cell migration

## Abstract

The involvement of the N-methyl-D-aspartate receptor (NMDAR), a glutamate-gated ion channel, in promoting the invasive growth of cancer cells is an area of ongoing investigation. Our previous findings revealed a physical interaction between NMDAR and MET, the hepatocyte growth factor (HGF) receptor. However, the molecular mechanisms underlying this NMDAR/MET interaction remain unclear. In this study, we demonstrate that the NMDAR2B subunit undergoes proteolytic processing, resulting in a low-molecular-weight form of 100 kDa. Interestingly, when the NMDAR2B and MET constructs were co-transfected, the full-size high-molecular-weight NMDAR2B form of 160 kDa was predominantly observed. The protection of NMDAR2B from cleavage was dependent on the kinase activity of MET. We provide the following evidence that MET opposes the autophagic lysosomal proteolysis of NMDAR2B: (i) MET decreased the protein levels of lysosomal cathepsins; (ii) treatment with either an inhibitor of autophagosome formation or the fusion of the autophagosome and lysosome elevated the proportion of the NMDAR2B protein’s uncleaved form; (iii) a specific mTOR inhibitor hindered the anti-autophagic effect of MET. Finally, we demonstrate that MET coopts NMDAR2B to augment cell migration. This implies that MET harnesses the functionality of NMDAR2B to enhance the ability of cancer cells to migrate.

## 1. Introduction

The ionotropic N-methyl-D-aspartate receptor (NMDAR) is a glutamate-gated ion channel that mediates signaling at the majority of excitatory synapses within the nervous system [1]. NMDARs are tetramers composed of the NMDAR1 subunit in complex with NMDAR2- and NMDAR3-type regulatory subunits [2,3]. Several lines of evidence indicate that the NMDAR2B subunit plays an important role in brain development, circuit formation, differentiation, and synaptic plasticity [4]. The importance of this subunit is emphasized by the neonatal lethality of NMDAR2B knock-out mice [5]. This highlights the vital role played by NMDAR2B in the overall functioning and survival of organisms. While NMDARs are traditionally associated with neuronal function, recent studies have demonstrated their presence and involvement in various cancer types [6,7,8,9,10,11,12,13,14]. Zeng et al. [15] identified the elevated expression of the NMDAR2B subunit in breast-to-brain metastasis. They found that tyrosine-phosphorylated NMDAR2B, which is crucial for the activation of cell-surface NMDARs [16,17,18], was detected at higher levels in breast-to-brain metastasis compared to primary breast tumors. The inhibition of NMDAR2B in these cells resulted in a reduced brain tumor size and increased survival in mice, indicating a potential role of NMDAR in promoting the growth of invasive cancer cells in the brain.

In a recent study [19], we provided evidence of a physical interaction between NMDAR and MET. *MET* encodes the tyrosine kinase receptor responsible for binding to hepatocyte growth factor (HGF), initiating signaling cascades that drive cell migration, tumor invasion, and metastasis [20]. Our research aims to uncover the molecular mechanisms that underlie the role of MET in facilitating its physical interaction with the glutamate receptor, shedding light on the signaling pathways implicated in this process. This knowledge may provide valuable insights into the development of targeted therapies aimed at disrupting this pathway and potentially inhibiting tumor metastasis.

## 2. Materials and Methods

### 2.1. Cell Culture and Materials

Hek293T cells and the human breast carcinoma TNBC BT-549 cell line were purchased from the American Type Culture Collection (ATCC, Manassas, VA, USA) and cultured in Iscove and RPMI, respectively. The media were added with 10% of FBS, 1% of penicillin, 1% of streptomycin, and 1% of L-glutamine. All reagents, unless specified, were from Sigma Aldrich (St. Louis, MO, USA). HGF (recombinant human HGF NS0-expressed) was purchased from R&D systems (Minneapolis, MN, USA). Cells were tested for mycoplasma one time per week. MET tyrosine kinase inhibitor JNJ-38877605 (JNJ) was purchased from Selleckchem (Houston, TX, USA). Bafilomycin A (Baf, vacuolar-type H+-ATPase), 3-methyladenine (3-MA, inhibitor of class III PI3K Vps34), and temsirolimus (Tem, mTOR inhibitor) were purchased from Sigma Aldrich.

### 2.2. Immunofluorescence Analysis

Hek293T cells were plated in fibronectin (6 µg/mL)-coated 24-well plates and fixed for 10 min with ice-cold 4% PBS paraformaldehyde (Santa Cruz Biotechnology, Dallas, TX, USA). A total of 0.1% of Triton X-100 and 1% of BSA was used for permeabilization and saturation, respectively. Then, the cells were incubated with a primary antibody (Appendix A) for 1 h at room temperature. Incubation with secondary antibodies was performed using a specific Alexa 488 (green) or 555 (red)-conjugated anti-goat/rabbit/mouse secondary antibody (Invitrogen, Waltham, MA, USA) for 1 h at room temperature. Duolink in situ mounting medium (Sigma Aldrich) with DAPI was used for nuclear staining and mounting. A TCS SP2 AOBS confocal laser-scanning microscope and the LAS AF v2.6 software (Leica Microsystems, Wetzlar, Germany) were used to obtain immunofluorescence images.

### 2.3. Proximity Ligation Assay (PLA)

Cells were plated and fixed as described in the “Immunofluorescence Analysis” (Section 2.2). For PLA, the Duolink In Situ Detection Reagents Orange kit (DUO92007, Sigma Aldrich) was used, following the manufacturer’s instructions. Anti-MET (homemade antibody), NMDAR2B (ab65783, Abcam, Cambridge, UK), and P-NMDAR2B (Tyr1252, 48-5200, Invitrogen) antibodies were exploited. The Duolink in situ mounting medium (Sigma Aldrich) with DAPI was used for nuclear staining and mounting. The TCS SP2 AOBS confocal laser-scanning microscope and LAS AF v2.6 software (Leica Microsystems) were used for fluorescence analysis.

### 2.4. Immunoprecipitation Assay

Cells were lysed in ice-cold RIPA buffer in the presence of a cocktail of protease inhibitors. Total protein lysates were incubated on rotor with anti-MET home-made antibodies or anti-Flag/His Tag antibodies overnight at 4 °C and then with sepharose protein A (GE Healthcare Systems, Chicago, IL, USA) for 2 h at 4 °C. Incubation with sepharose protein A in the absence of antibodies was used as a control. Subsequently, five washes with ice-cold RIPA buffer and elution with boiling Laemmli buffer were performed. Immunoprecipitated proteins were separated on SDS-PAGE and analyzed by Western blotting (for primary antibodies, see Appendix A).

### 2.5. Western Blot Analyses

Cells were lysed in ice-cold RIPA added with a protease inhibitor cocktail. Lysates were subsequently sonicated and centrifuged at 12,000× *g* at +4 °C for 20 min. The BCA Protein Assay Kit (Thermo Fisher Scientific, Waltham, MA, USA) was used to evaluate the protein concentration. Normalized protein lysates were separated by electrophoresis together with a prestained protein ladder (10–180 kDa, PageRuler, Thermo Fisher Scientific) on 4–12% precast gels for SDS-PAGE (Invitrogen). After gel running, proteins were transferred to a Hybond-P pvdf membrane (Bio-Rad, Hercules, CA, USA). The membranes were blocked with 10% of BSA at room temperature and subsequently incubated with primary antibodies (Appendix A) overnight at +4 °C, and with specific HRP-conjugated secondary antibodies (Jackson Laboratory, Bar Harbor, ME, USA) for 1 h at room temperature. The ECL Prime detection kit and Image Lab v5.2.1 software (Bio-Rad) were used for protein detection and quantification, respectively.

### 2.6. Plasmids

Codon optimization, the synthesis of NMDAR2B (GenBank: NM_000834.5) and MET (GenBank: NM_000245.4) expressing constructs, and mutagenesis for the MET kinase-dead (KD) mutant (K1110A) were commercially performed by GenScript (Piscataway, NJ, USA). NMDAR2B and MET cDNA-optimized sequences were inserted into pcDNA3.1+C-DYK (Flag) and the pcDNA3.1(+)-myc-His A plasmid, respectively. Plasmids (1 µg) were used for the transient transfection of Hek293T and BT-549 cells using Lipofectamine 2000 reagent (Thermo Fisher Scientific).

### 2.7. Cancer-Related Protein Array

The Human XL Oncology Array (ARY026, R&D), which allowed us to analyze 84 cancer-related proteins simultaneously, was exploited. To perform the assay, we followed the manufacturer’s protocol.

### 2.8. Wound-Healing Assay

Transfected cells were plated in 24-well plates (1,000,000 and 500,000 cells/well for Hek293T and BT-549, respectively) and maintained in culture until confluence. The monolayers were wounded with a plastic pipette. Images of wounds at the starting point and after 24 h were taken with a DMRI Leica inverted microscope. Migration was quantified by evaluating the area of the wound at time zero (A0) and after 24 h (Ay). Normalization and quantification on the basis of three independent experiments were performed. Areas were quantified with the ImageJ v1.50b software.

### 2.9. Statistical Analysis

All values are expressed as the mean ± standard deviation (S.D.). Immunofluorescence quantification was performed using the ImageJ software (number of replicates = 3). Yellow/red fluorescence was quantified and normalized to the nuclei number (DAPI staining). Statistical analysis was performed blindly on groups with a sample size of at least 3. A *t*-test was used to statistically compare two groups. For multiple comparisons, one-way ANOVA was used, followed by the Tukey post hoc test (*t*-test). The post hoc test was performed only if the F was significant and there was no variance in homogeneity. The threshold *p*-value deemed to constitute statistical significance was <0.05, and only data characterized by *p*-values < 0.05 are denoted throughout the paper as results with statistical significance. In the statistical analysis, all the samples were analyzed, and the outliers were not excluded. The data analysis and the graph design were performed using the GraphPad Prism v.8 software (GraphPad Software).

## 3. Results

### 3.1. Activated MET Physically Interacts with the Endogenous Full-Size NMDAR2B in Hek293T Cells

In our investigation, we examined the physical interaction between NMDAR2B and MET receptors in Hek293T cells. Utilizing confocal immunofluorescence analysis, we observed the colocalization of MET and NMDAR2B, as well as phosphorylated NMDAR2B at Tyr1252, indicated by merged staining and a distinct yellow signal (Figure 1a). HGF induced NMDAR2B phosphorylation preferentially in a subset of cells (Figure 1a and Appendix A). Notably, we observed a robust colocalization signal between HGF-activated MET and phosphorylated NMDAR2B (Figure 1a). To further confirm the close interaction, we employed PLA, which demonstrated a significant increase in the bright red fluorescent PLA dots following HGF treatment, indicating a strong interaction between the two molecules (Figure 1b). Coimmunoprecipitation experiments revealed that the pull-down of HGF-activated MET resulted in the enrichment of NMDAR2B phosphorylated at Tyr1252 (Figure 1c). The protein complex formed by phosphorylated NMDAR2B and activated MET exhibited a high molecular weight of 160 kDa (Figure 1c), corresponding to the full-size NMDAR2B. In contrast, the NMDAR2B band detected in the input lysate using the same anti-NMDAR2B C-terminal antibody showed a molecular weight of 100 kDa (Figure 1c).

### 3.2. Activated MET Opposes NMDAR2B Cleavage and Preferentially Interacts with the Uncleaved Form

In our experimental approach, we performed immunoprecipitation analysis using Hek293T cells transfected with pcDNA3 constructs expressing either NMDAR2B with a Flag tag or MET wild type with a His tag, individually and in combination. As depicted in Figure 2a, the anti-Flag Tag antibody successfully pulled down the 100 kDa form in NMDAR2B-transfected cells, while, in NMDAR2B+MET co-transfected cells, it captured the 160 kDa form along with MET (Figure 2a). In the latter cells, the anti-His Tag antibody specifically immunoprecipitated MET and the uncleaved 160 kDa form (Figure 2a). Furthermore, the Western blot analysis of the total lysates using anti-NMDAR2B antibodies exhibited a transition from the low-molecular-weight band to the high-molecular-weight band in the presence of MET (Figure 2b). Notably, the N-terminal antibody detected the 160 kDa band only in NMDAR2B+MET co-transfected cells. In contrast, the C-terminal antibody recognized both the 160 kDa form in NMDAR2B+MET co-transfected cells and the 100 kDa form in NMDAR2B-transfected cells, indicating that the latter likely corresponded to the C-terminal fragment of NMDAR2B (Figure 2b). Additionally, the ectopically expressed MET demonstrated significant autophosphorylation levels at Y1234–Y1235 due to the abundant protein synthesis resulting from the optimized ectopic transfected gene (Figure 2b). Confocal immunofluorescence analysis confirmed that the transfected NMDAR2B and MET proteins colocalized, as indicated by merged staining and a distinct yellow signal obtained with either the epitope tag (anti-Flag/His) or anti-N-terminal NMDAR2B/MET antibodies (Figure 2c and Appendix A). Overall, these findings strongly suggest that activated MET physically interacts with the uncleaved 160 kDa form of NMDAR2B, thereby preventing its cleavage and degradation at the N-terminus.

**Figure 1 cells-13-00028-f001:**
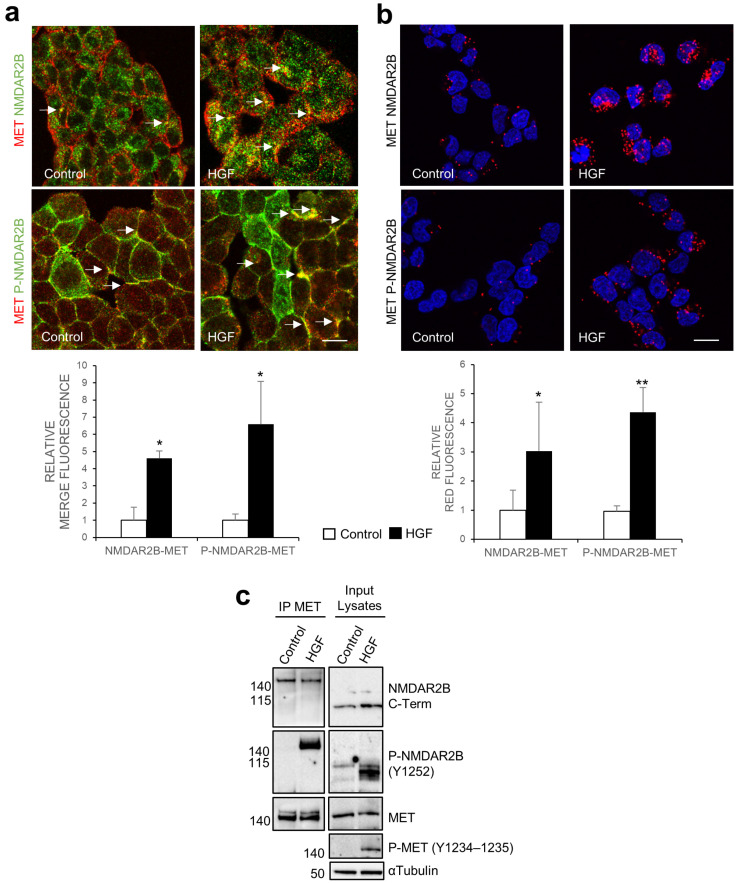
MET interacts with the endogenous full-size NMDAR2B in Hek293T cells. Hek293T cells were unstimulated (control) or stimulated with HGF (0.6 nM). (**a**) MET and NMDAR2B immunofluorescence co-staining (yellow, white arrows) was produced for MET (red) with total (upper panels) or phosphorylated (Tyr1252, lower panels) NMDAR2B (green) proteins. (**b**) PLA was performed using antibodies recognizing MET and total (upper panels) or phosphorylated (Tyr1252, lower panels) NMDAR2B proteins. Red fluorescent profiles represent regions of PLA signal amplification, denoting MET and NMDAR physical proximity. DAPI staining was used to detect nuclei (blue). Representative confocal microscopy images of fluorescence/PLA and quantitative analysis of MET/NMDAR signals are shown. Bar = 50 µm. Values are the mean ± S.D. of three independent experiments. A *t*-test was applied for comparison of HGF-treated versus untreated cells. ** *p*-value < 0.01; * *p*-value < 0.05. (**c**) MET IP and input lysates were analyzed by Western blot with total C-terminal NMDAR2B, phosphorylated NMDAR2B (Tyr1252), total MET, and phosphorylated MET (Tyr1234-1235) antibodies. α-Tubulin was used as a loading control.

**Figure 2 cells-13-00028-f002:**
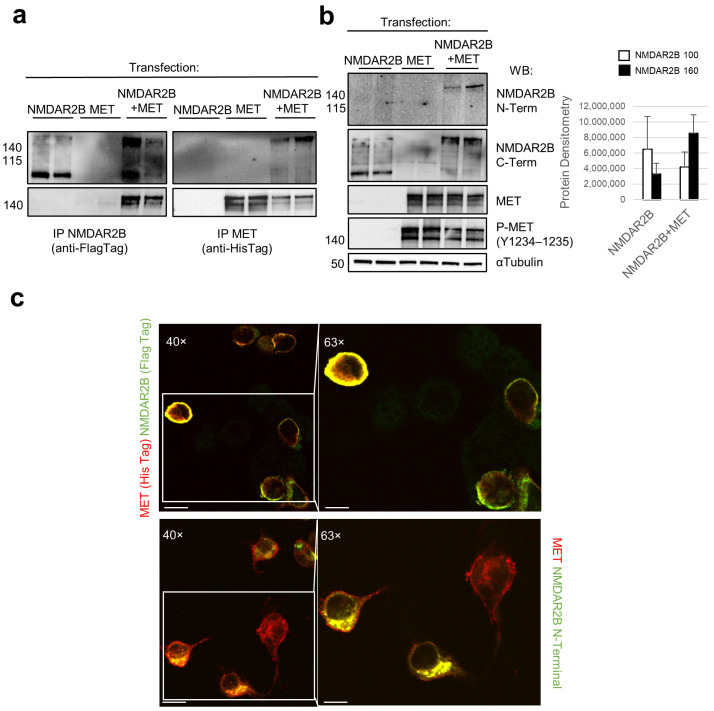
Activated MET counteracts NMDAR2B cleavage and preferentially interacts with the uncleaved form. Hek293T cells were transfected for 24 h with pcDNA3 constructs expressing NMDAR2B with a Flag tag or MET wild type with a His tag, individually and in combination. Anti-Flag/anti-His Tag IP (**a**) and input lysates (**b**) were analyzed by Western blot with C-terminal NMDAR2B and MET antibodies. Input lysates were also decorated with N-terminal NMDAR2B and phosphorylated MET (Tyr1234–1235) antibodies. α-Tubulin was used as a loading control. Densitometric analysis for the 100 and 160 kDa forms of NMDAR2B was performed on input lysates with Western blot. Data are representative results of three independent experimental replicates. (**c**) MET and NMDAR2B immunofluorescence co-staining (yellow) was evaluated with anti-Flag (upper panels, NMDAR2B, green)/His (MET, red) and anti-N-terminal NMDAR2B (lower panels, green)/MET (red) couple of antibodies. Images were taken by two enlargements (40 and 63×). Bar 40× = 50 µm, Bar 63× = 80 µm.

### 3.3. The MET Inhibitor and the MET Kinase-Dead Mutant Diminish the Protective Effect of MET on the Cleavage and Degradation of NMDAR2B

Upon the treatment of NMDAR2B and MET co-transfected cells with the MET tyrosine kinase inhibitor JNJ-38877605 (JNJ), the full-size band of NMDAR2B was reduced, and the C-terminal fragment became more prominent (Figure 3a). This observation indicates that the protection of NMDAR2B from cleavage, mediated by MET, was compromised by blocking the tyrosine kinase enzymatic activity. In addition, co-immunoprecipitation experiments conducted on cells co-transfected with NMDAR2B and the MET kinase inactive receptor, hereafter referred to as MET KD (MET kinase-dead), revealed a substantial increase in the abundance of the 100 kDa fragment, with no detection of either NMDAR2B band in the anti-His Tag (MET KD) pull-down (Figure 3b,c). These results collectively suggest that the kinase activity of MET is essential for the physical interaction between NMDAR2B and MET, as well as for the protection against NMDAR2B protein cleavage and subsequent degradation.

**Figure 3 cells-13-00028-f003:**
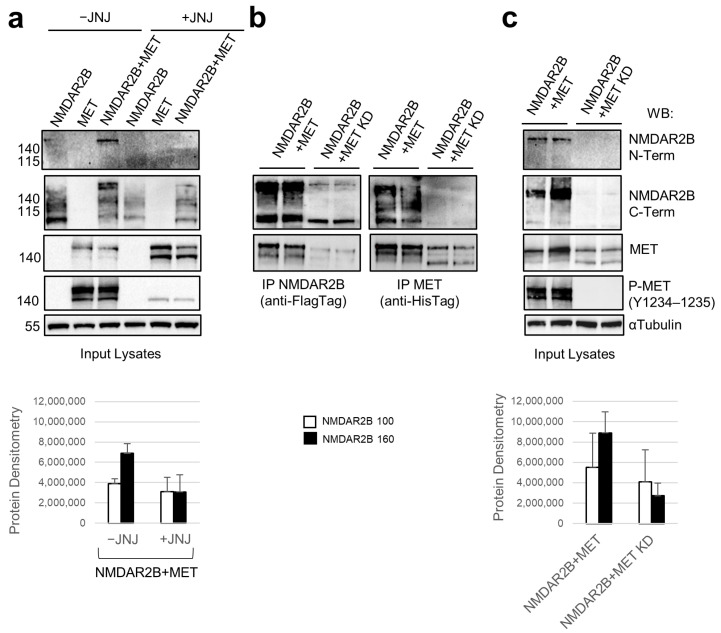
Inhibition of MET kinase activity quenches its protection against NMDAR2B cleavage. Hek293T cells were transfected for 24 h with pcDNA3 constructs expressing NMDAR2B with a Flag tag and MET wild type or MET kinase-dead mutant (MET KD) with a His tag, individually and in combination. (**a**) Transfected cells were treated with the specific MET tyrosine kinase inhibitor JNJ-38877605 (JNJ, 500 nM) or untreated (-JNJ). JNJ was added concomitantly with the transfection. Input lysates (**a**,**c**) and anti-Flag/anti-His Tag IP (**b**) were analyzed by Western blot with total C-terminal NMDAR2B and MET antibodies. Input lysates were also decorated with total N-terminal NMDAR2B and phosphorylated MET (Tyr1234–1235) antibodies. α-Tubulin was used as a loading control. Densitometric analysis for the 100 and 160 kDa forms of NMDAR2B was performed on input lysates with Western blot. Data are representative results of three independent experimental replicates.

### 3.4. MET Safeguards against NMDAR2B Cleavage by Inhibiting Autophagy through the mTOR Pathway

In our efforts to uncover the molecular mechanisms underlying the MET-mediated protection from NMDAR2B cleavage, we performed proteomic array screening, simultaneously examining 84 human cancer-related proteins, including proteases and protease inhibitors frequently dysregulated in cancer (Appendix A). Hek293T cells were either untreated or transfected with the MET-expressing construct, and the protein samples were probed with the array to evaluate the relative protein levels (Figure 4a). Compared to the untreated samples, MET-transfected cells exhibited reduced levels of Cathepsin B and D (Figure 4a). Cathepsin B and D are proteases primarily localized in the lysosome. They are involved in the degradation of endocytosed extracellular proteins, as well as the turnover of intracellular proteins. They exhibit optimal activity at an acidic pH, which aligns with the lysosomal environment, and are implicated in various cellular processes, including autophagy. In order to investigate the lysosomal targeting of NMDAR2B via autophagic cargo, we employed two selective inhibitors. 3-methyladenine (3-MA) is a selective inhibitor of the autophagic initiator PI3K type III. Bafilomycin A (Baf) selectively inhibits the activity of the vacuolar-type H+-ATPase (V-ATPase), which is responsible for lysosome acidification and fusion between autophagosomes and lysosomes, thus inhibiting the degradation of autophagic cargo. An important phase of autophagy is the elongation of the autophagosome, with the generation of the active lipidated form of LC3, called LC3II, from the inactive LC3I form [21]. The treatment of Hek293T cells with 3-MA increased the LC3I/LC3II ratio (LC3I increased relatively to LC3II, Figure 4b) resulting from the 3-MA-dependent blockage of autophagy initiation. Moreover, Baf treatment significantly decreased the LC3I/LC3II ratio (LC3II increased relatively to LC3I, Figure 4b), indicating the blockage of the basal lysosomal-dependent degradation of the autophagosome cargo. Consequently, we noted an elevation in the proportion of the NMDAR2B protein’s uncleaved form relative to its cleaved form following either treatment (Figure 4b). In previous work, we demonstrated that MET decreases the autophagic flux through the activation of mTOR, which is known to be the main inhibitor of autophagy [22]. Indeed, the activation of the mTOR pathway was evident in Hek293T cells transfected with MET. This was demonstrated by the elevated levels of phospho-p70S6K, a widely recognized substrate of mTOR activation (Figure 4c). The protective effect of MET against NMDAR2B cleavage was blunted by temsirolimus (Tem), a specific mTOR inhibitor and inducer of autophagy (Figure 4c). In fact, Hek293T cells co-transfected with NMDAR2B and MET and treated with Tem showed increased levels of LC3II relative to LC3I, as proof of autophagy induction; a lack of p70S6K phosphorylation, as a consequence of mTOR inhibition; and the absence of the 160 kDa NMDAR2B form (Figure 4c). Collectively, these findings suggest that the cleavage of NMDAR2B takes place through the degradation of the autophagic cargo in the lysosome, and MET protects NMDAR2B from cleavage by decreasing the autophagic flux via mTOR signaling.

### 3.5. MET Coopts Uncleaved NMDAR2B to Augment Cell Migration

Since *MET* is a known master regulator gene for cell migration, we tested the influence of the NMDAR–MET complex on cell migration (Figure 5). We performed the wound-healing assay on Hek293T cells transfected with different constructs, including NMDAR2B alone, MET alone, and both NMDAR2B and MET together. The results showed that cells co-transfected with NMDAR2B and MET exhibited higher migration levels compared to cells transfected with NMDAR2B or MET alone (Figure 5a). This indicates that the presence of both NMDAR2B and MET enhances cell migration. We extended the study to BT-549 cells, a type of triple-negative breast carcinoma. In these cells, coimmunoprecipitation experiments confirmed that the protein complex formed by NMDAR2B and MET exhibited a high molecular weight of 160 kDa (Figure 5b), corresponding to the full-size NMDAR2B. In contrast, the NMDAR2B band detected in the input lysate using the same anti-NMDAR2B C-terminal antibody showed a molecular weight of 100 kDa (Figure 5b). HGF-activated MET resulted in the enrichment of NMDAR2B phosphorylated at Tyr1252 and protein complex formation (Figure 5b). Moreover, we transfected pcDNA3 constructs expressing NMDAR2B with a Flag tag or MET wild type with a His tag, both individually and in combination. We observed that the 160 kDa form was detected only in NMDAR2B+MET co-transfected cells (Figure 5c). Additionally, the transfection of the MET-expressing construct led to receptor autophosphorylation (Y1234–Y1235), indicating the activation of the MET receptor (Figure 5c). These findings in BT-549 cells confirmed what was observed in Hek293T cells and suggested that activated MET protects NMDAR2B from cleavage. Finally, to validate the hypothesis that MET coopts NMDAR2B in the pro-migratory cell program, the wound-healing assay was performed on BT-549 cells co-transfected with NMDAR2B and MET. The results demonstrated a significant increase in cancer cell migration over the wound, providing further evidence that MET collaborates with NMDAR2B to promote cell migration (Figure 5d).

**Figure 4 cells-13-00028-f004:**
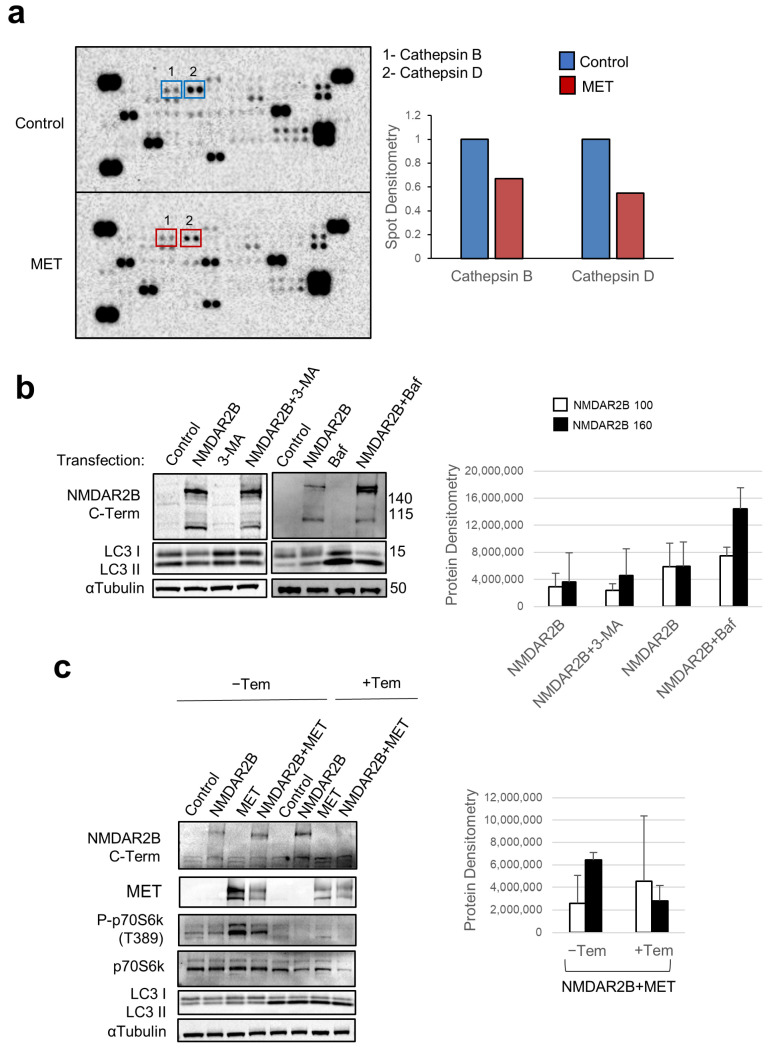
MET safeguards against NMDAR2B cleavage by inhibiting autophagy through mTOR pathway. (**a**) Protein samples obtained from Hek293T cells transfected for 24 h with pcDNA3 construct expressing MET wild type with a His tag were probed with the Human XL Oncology Array. Densitometric analysis of lysosomal proteins, Cathepsin B and D, was performed. (**b**) Hek293T cells transfected for 24 h with pcDNA3 construct expressing NMDAR2B with a Flag tag were untreated (control) or treated with 3-methyladenine (3-MA, 1 mM), a selective inhibitor of the autophagic initiator PI3K type III, or Bafilomycin A (Baf, 50 nM), an inhibitor of the vacuolar-type H+-ATPase and the fusion between the autophagosome and lysosome. 3-MA and Baf were added concomitantly with the transfection. Total C-terminal NMDAR2B and LC3I/II proteins were analyzed by Western blot. (**c**) Hek293T cells transfected for 24 h with pcDNA3 constructs expressing NMDAR2B with a Flag tag or MET wild type with a His tag, individually and in combination, were treated with temsirolimus (+Tem, 1 µM), a specific mTOR inhibitor, or untreated (-Tem). Tem was added concomitantly with the transfection. Total C-terminal NMDAR2B, total MET, total and phosphorylated (Thr389) p70S6K, and LC3I/II proteins were analyzed by Western blot. In (**b**,**c**), α-Tubulin was used as a loading control, densitometric analyses for the 100 and 160 kDa forms of NMDAR2B were performed, and data are representative results of three independent experimental replicates.

**Figure 5 cells-13-00028-f005:**
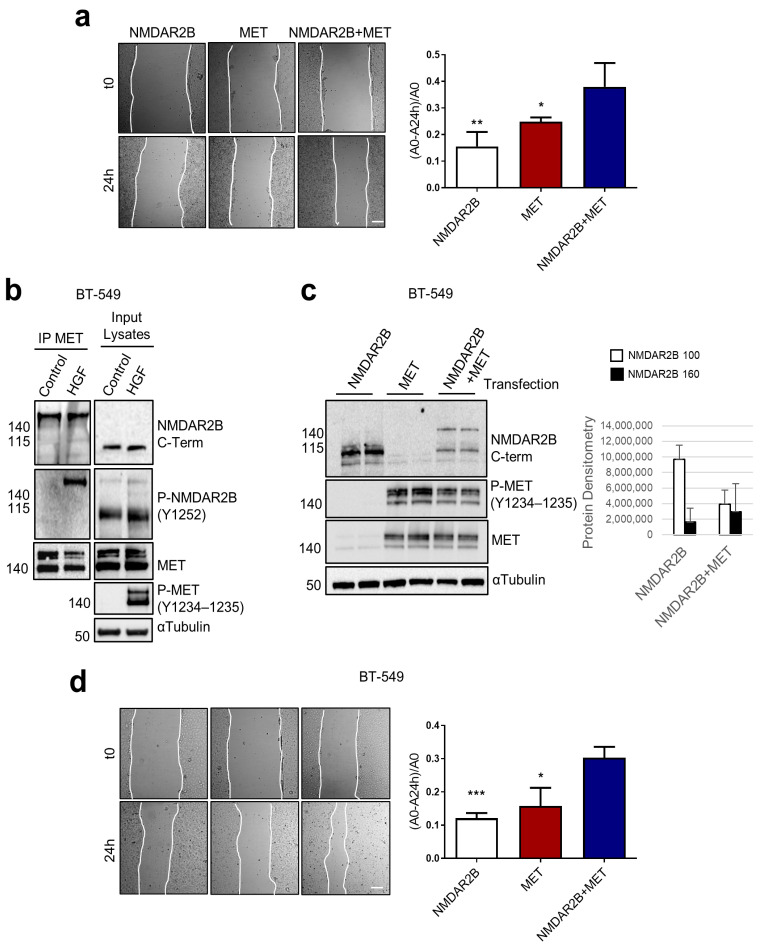
MET coopts uncleaved NMDAR2B to induce cell migration. (**a**) Hek293T cells were transfected with pcDNA3 constructs expressing NMDAR2B with a Flag tag or MET wild type with a His tag, individually and in combination. The promigratory function was assayed by wound-healing assays. Representative images and wound closure quantification are reported. Bar = 300 µm. Migration was quantified by evaluating the area of the wound at time zero (A0) and after 24 h (Ay). Values are the mean ± S.D. of three independent experiments. A *t*-test was applied to compare each sample versus NMDA2B+MET-transfected cells. *** *p*-value < 0.001; ** *p*-value < 0.01; * *p*-value < 0.05. (**b**) TNBC BT-549 cells were unstimulated (control) or stimulated with HGF (0.6 nM). MET IP and input lysates were analyzed by Western blot with total C-terminal NMDAR2B, phosphorylated NMDAR2B (Tyr1252), total MET, and phosphorylated MET (Tyr1234-1235) antibodies. α-Tubulin was used as a loading control. (**c**,**d**) TNBC BT-549 were transfected as Hek293T (**a**). (**c**) Total C-terminal NMDAR2B, phosphorylated (Tyr1234-1235), and total MET were analyzed by Western blot. α-Tubulin was used as a loading control. Densitometric analysis for the 100 and 160 kDa forms of NMDAR2B was performed and data are representative results of three independent experimental replicates. (**d**) The promigratory function was assayed by wound-healing in BT-549 as in Hek293T cells (**a**).

In conclusion, this study reveals that the interaction between MET and NMDAR2B influences cell migration, with activated MET preventing the cleavage of NMDAR2B and stimulating a pro-migratory cell program. These findings contribute to a better understanding of the molecular mechanisms underlying cell migration and may have implications for cancer research and therapeutic interventions.

## 4. Discussion

The interaction between the NMDAR, a key player in synaptic plasticity and learning, and the MET receptor has garnered recent scientific interest in the context of neuron wiring and synaptogenesis [23]. Studies have shown that NMDAR function and signaling can be regulated by MET activation, influencing neuronal development, synaptic transmission, and plasticity [24,25,26]. This interaction may have implications for neurodevelopmental disorders and neurodegenerative diseases, providing potential targets for therapeutic interventions aimed at modulating the NMDAR/MET interaction [27,28]. Due to the intriguing correlation observed between NMDAR and MET in neurons, our previous investigation aimed to establish a connection between NMDAR and the HGF–MET pathway in the context of non-neuron cells such as breast cancer cells [19]. We sought to explore the potential relationship between these two signaling systems to better comprehend their interplay and significance in cancer biology. The findings of the present study provide compelling evidence that a physical interaction between NMDAR and MET occurs also in other non-neuronal cells, such as Hek293T cells. The kinase activity of MET is necessary for this interaction to occur and is crucial for safeguarding NMDAR2B from protein cleavage and subsequent degradation. A small subset of cells exhibits NMDAR2B phosphorylation in response to HGF, leading to protection from cleavage. This occurrence is attributed to the stochastic nature of the interaction between the two receptors, necessitating their close proximity at the plasma membrane and the phosphorylation of both receptors to be stabilized. These results highlight the regulatory role of MET kinase activity in preserving the integrity and stability of the NMDAR2B protein, emphasizing its importance in modulating NMDAR-related signaling pathways and cellular functions. During our investigation into the molecular mechanisms responsible for MET-mediated protection against NMDAR2B cleavage, we made a significant discovery that MET mitigates autophagy through the activation of the mTOR pathway, a known suppressor of the initiation and progression of autophagy. By inhibiting autophagy through mTOR activation, MET helps to protect cells from undergoing NMDAR2B cleavage.

The involvement of autophagy in the proteolytic cleavage of NMDAR in cancer cells is not surprising. Many types of cancer exhibit high autophagic activity [29]. In certain contexts, autophagy can promote cancer progression. In established tumors, autophagy can facilitate tumor cell survival and adaptation to harsh conditions, such as nutrient deprivation and hypoxia. It allows cancer cells to recycle cellular components, providing the necessary building blocks for sustained growth and proliferation. Glutamine and glutamate are critical amino acids that can supply carbon and nitrogen for proteins, nucleotides, and lipid synthesis [30,31,32,33]. Glutamine can be converted by glutaminase into an ammonium ion and glutamate. Ammonia, released in the microenvironment, is a paracrine and autocrine inducer of autophagy, and thus triggers a positive feedback signal that further promotes autophagy [34]. Through subsequent reactions catalyzed by glutamate dehydrogenase, glutamate can become alpha-ketoglutarate and fuel the tricarboxylic acid cycle (TCA) to produce ATP. This conversion can also be carried out by aminotransferases, thus contributing to the synthesis of non-essential amino acids for the cellular biomass [35]. In addition to being a bioenergy and biomass substrate for cancer cells, a proportion of the glutamate pool is destined for the extracellular milieu. The xCT antiporter facilitates the exchange of glutamate for cystine, which is rapidly converted to cysteine within the cell. This process contributes to the production of glutathione, enabling the mitigation of reactive oxygen species (ROS) and the alleviation of oxidative stress in cancer cells [36]. The accumulation of glutamate in the extracellular milieu can lead to the overstimulation of its membrane receptors, such as NMDAR, leading to cell injury or death, a condition known as excitotoxicity in neurons [37]. We show that NMDAR2B, the subunit that binds glutamate, is cleaved at the N-terminus in Hek293T cells and BT-549 TNBC cells. The NMDAR2B cleavage by cancer cells may be an adaptive mechanism to avoid excitotoxic cell death. The proteolytic cleavage of NMDAR through autophagy can contribute to the downregulation of surface receptor levels and synaptic function. We suggest that by preventing the cleavage of NMDAR2B, MET helps to maintain its stability and functional integrity, thereby preserving its functional role. Khamsing D et al. [38] observed, in mouse hippocampal neurons, that the brain-derived neurotrophic factor (BDNF) synergistically worked together with NMDAR, powering the two arms of mTORC1 activation: mTORC1 translocation to the lysosome and Rheb activation. The mTOR pathway, in turn, suppresses the autophagic pathway. Accordingly, rapastinel, a novel glutamatergic glycine-like NMDAR partial agonist, activates BDNF/mTOR signaling [39]. This NMDAR2B-dependent mechanism could create a positive loop with MET, avoiding its own cleavage and enhancing pro-migratory functions. Synapses, which are specialized junctions between neurons facilitating neuronal communication, are typically associated with the nervous system. However, it has been discovered that cancer cells can exhibit synaptic-like properties and engage in synapse-like interactions with neighboring cells or components of the tumor microenvironment [40]. Synaptic proteins, such as NMDARs, can modulate cancer cell behavior. These synaptic proteins may contribute to cell adhesion, migration, invasion, and even the formation of tumor cell networks resembling neural circuits, as suggested by Zeng et al. [15] for breast-to-brain tumors. These synapse-like structures, referred to as “tumor synapses” or “oncogenic synapses”, involve the formation of specialized contacts among cancer cells and between cancer cells and immune cells, stromal cells, or even neural cells in the tumor microenvironment. These synapse-like interactions can facilitate communication, signal transmission, and molecular transfer, thereby influencing tumor growth, invasion, immune evasion, and metastasis.

While the precise mechanisms and functional implications of synaptic-like functions in cancer are still being actively researched, our findings highlight a previously unrecognized aspect of cancer biology and suggest that targeting synaptic interactions in the tumor could hold therapeutic potential. Further investigations are needed to unravel the complex interplay between synapse-like structures, cancer cells, and the tumor microenvironment to fully understand the impact of synaptic functions on cancer progression and therapeutic strategies.

## 5. Conclusions

In summary, our investigation into the interplay between the NMDAR and MET receptors extends beyond the realm of neuronal function, reaching into the context of non-neuronal cells, particularly in breast cancer cells. The observed physical interaction between NMDAR and MET in Hek293T cells underscores the versatility of this relationship beyond the neural context. Crucially, the kinase activity of MET emerges as a key determinant for safeguarding NMDAR2B from cleavage. Moreover, our study reveals a novel connection between MET and autophagy through the activation of the mTOR pathway. By inhibiting autophagy, MET protects cells from NMDAR2B cleavage, suggesting a potential mechanism by which MET contributes to cancer cell survival and adaptation to challenging conditions. The link between NMDAR2B cleavage and autophagy in cancer cells adds a layer of complexity to our understanding of cancer biology. The adaptive nature of NMDAR2B cleavage in response to autophagy may represent a strategy employed by cancer cells to avoid excitotoxic cell death. Our findings suggest that MET’s role in preventing NMDAR2B cleavage contributes to maintaining the stability and functional role of NMDAR2B.

The potential positive loop involving MET, NMDAR2B, and mTOR signaling, as well as the implications of these interactions in promoting pro-migratory functions, introduces a novel dimension to our understanding of cancer progression, paving the way for future research to explore the therapeutic possibilities arising from targeting these signaling pathways in cancer.

## Data Availability

The data presented in this study are available in the article and in Appendix A.

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
