# Peer review of "MET Oncogene Enhances Pro-Migratory Functions by Counteracting NMDAR2B Cleavage"

_cells, 2023, doi:10.3390/cells13010028_

Round 1

Reviewer 1 Report (Previous Reviewer 2)

Comments and Suggestions for Authors

The authors addressed all the questions and comments of the reviewer in a convincing manner. 

Reviewer 2 Report (Previous Reviewer 1)

Comments and Suggestions for Authors

NO further question

This manuscript is a resubmission of an earlier submission. The following is a list of the peer review reports and author responses from that submission.

Round 1

Reviewer 1 Report

Comments and Suggestions for Authors

1. Why the p-NMDAR2B is higher than total NMDAR2B in Figure 1

2. In Figure 2a IP MET, what is the band spanning the last two lanes?

3. In all confocal images, both separated and merged images should be provided

4. If NMDAR2B is involved, what is the mechanism without glutamate? Experiments with NMDAR2B agonist and antagonist in neuron may be helpful.

Minor concerns:

1. The abbreviations should be given full name and used consistently in subsequent context, for example, NMDAR HGF, PLA.

Comments on the Quality of English Language

The language need to be improved.

Reviewer 2 Report

Comments and Suggestions for Authors

This manuscript of Gallo et al. is a follow-up study of a paper on the interplay between the receptor tyrosine kinase MET and the glutamate-gated ion channel subunit NMDAR2B published in Cancers last year. There, the physical interaction between MET and NMDR2B has been shown in two triple-negative breast cancer cell lines. In this study, the appearance and function of the uncleaved 160 kDa NMDAR2B protein and its 100 kDa C-terminal cleavage product in respect to the expression of active MET were analyzed in detail. Co-expression of active MET in HEK293 cells and breast cancer BT-549 cells protected NMDAR2B from cleavage correlating with enhanced cell migration. The novelty of this manuscript is that NMDR2B degradation is connected to the autophagic lysosomal proteolysis of NMDAR2B, an effect that is opposed by active MET. Using a Human XL Oncology Array the authors identified a decrease of the lysosomal proteases Cathepsin B and D in MET overexpressing cells. The use of lysosomal inhibitors 3-Methyalanine and Bafilomycin A as well as the mTOR inhibitor and autophagy inducer temsirolimus supported the notion that Met protects NMDAR2B from cleavage by autophagic processes. Mostly, the data are convincing and clearly presented. However, some issues should be addressed.

Fig. 1a: What is the reason that the staining of endogenous NMDAR2B and MET is not equally distributed in HEK293 cells. Why did HGF induce the phosphorylation of endogenous NMDAR2B preferentially in a subset of HEK293 cells?

Fig. 1c: Can the uncleaved phosphorylated NMDAR2B protein be detected in HGF-treated cell lysates upon longer exposure time. Is it correct, that only a small subfraction of NMDAR2B is phosphorylated in an HGF-dependent manner and protected from cleavage?

Fig. 4a: Catephsin should be changed to Cathepsin

Fig. 5b: Triple-negative breast BT-549 expresses endogenous MET and NMDAR2B. To complete the data set, the authors should show an experiment with BT-549 cells treated with and without HGF to demonstrate the effect of HGF-induced phosphorylation of NMDAR2B correlating with its protection from cleavage similar to the data shown for HEK293T in Fig 1c.

Figures should be integrated into the main text.